# Physical and Psychological Benefits of a 12-Week Zumba Gold^®^ Exercise Intervention in Postmenopausal Sedentary Women from Low Socioeconomic Status

**DOI:** 10.3390/healthcare13172250

**Published:** 2025-09-08

**Authors:** Anne Delextrat, Alba Solera-Sanchez, Emma L. Davies, Sarah E. Hennelly, Clare D. Shaw, Lily Sabir, Adam Bibbey

**Affiliations:** 1School of Sport, Nutrition and Allied Health Professions, Oxford Brookes University, Oxford OX3 8HU, UK; asolera-sanchez@brookes.ac.uk (A.S.-S.); clareshaw@brookes.ac.uk (C.D.S.); abibbey@brookes.ac.uk (A.B.); 2The Centre for Psychological Research, Oxford Brookes University, Oxford OX3 8HU, UK; edavies@brookes.ac.uk (E.L.D.); shennelly@brookes.ac.uk (S.E.H.); lsabir@brookes.ac.uk (L.S.)

**Keywords:** exercise, dance, lower limb strength, endurance, mental health, motivation, social connectedness

## Abstract

Background: Exercise is known to help with the effects of the menopause, but women from low socioeconomic status (SES) tend to exercise less at this stage of life. Therefore, this study aimed to investigate the physical and psychological effects of a Zumba Gold^®^ community-based intervention in postmenopausal women from low SES. Methods: Forty-three women were divided into a Zumba Gold^®^ (ZG) and control (C) group, and participated in pre- and post-testing sessions, separated by a 12-week intervention period. The outcomes measured were: body composition, sit-to-stand (STS), six-minute walk test (6MWT), balance, Short Form Health Survey (SF-12), Multidimensional Fatigue Index (MFI-20), Self-Efficacy for Exercise (SEE), Behavioral Regulations in Exercise (BREQ-2), Hospital Anxiety and Depression Scale (HADS), and Social Connectedness questionnaires. Semi-structured interviews were also performed on 15 ZG participants. Results: The results showed significant improvements in the ZG group only for STS, 6MWT, mental health, fatigue (general, physical, and mental), social connectedness score, and a significantly lower level of amotivation. Zumba Gold^®^ had positive effects on four main themes: belonging, non-judgmental place, psychological motivational factors, and mind–body connection. Conclusions:This is the first study to show that ZG is beneficial for this population and encourages similar studies on other aspects of the menopause.

## 1. Introduction

During the menopause transition, women experience various physiological and psychological changes, some of which persist into older age [1,2]. As a result, older women are usually characterized by a redistribution and increase of fat mass, a decrease in muscle mass (i.e., sarcopenia), and a decline in psychological and cognitive function [3,4]. These changes can drastically affect women’s overall health. Indeed, sarcopenia can lead to a decrease in lower limb strength, functional capacity, walking speed, and impaired balance, thus increasing the risk of falls. In addition, obesity and individuals who are overweight have been associated with high blood pressure, a greater risk of cardiovascular disease, and some cancers [1,5]. In addition, a recent review highlighted that anxiety and depression were common causes of concern in postmenopausal women [5].

It is well-known that physical exercise is beneficial for older adults to increase muscle mass, decrease body fat and improve mental health, and reduce physical and mental fatigue [6]. However, a large number of women decrease their physical activity levels around the menopause, for various reasons, including lack of time or motivation, and physical symptoms, such as joint pain and fatigue [7]. This increased sedentariness is more important in older women from lower socioeconomic status (SES) who are twice as likely to be inactive as older women from high SES [8]. A study including 63 older adults, including 40 women, showed that neighborhood SES was one of the main determinants of decreased leisure-time physical activity in this population [9]. Despite these rather alarming results, older adults from low SES have not been the object of many scientific studies on physical activity [10]. Interestingly, while older women from various SES share similar barriers to physical activity (proximity of sports facilities, physical symptoms, body image, and intrinsic motivation linked to exercise type), there are some barriers, such as financial cost and accountability to others/social interactions, that have been shown to be specific to older women from lower SES [10,11]. In view of these barriers, it is important to consider motivational factors in the design of exercise interventions in this population. The self-determination theory (SDT) [12] provides a framework to understand motivation to exercise and suggests that individuals may exhibit different regulations, namely, amotivation (lack of motivation), extrinsic, and intrinsic motivation. It proposes that individuals have three basic psychological needs: autonomy, competence, and relatedness [12]. The greater satisfaction of these needs leads to greater internalization and integration, which promotes a shift from amotivation, through external (i.e., external demands such as reward), introjected (i.e., contingent self-esteem, guilt), identified (i.e., consciously valued goals), and integrated (i.e., values fully assimilated into self) regulations, ultimately leading to intrinsic motivation (i.e., taking part for enjoyment, challenge, and curiosity). A systematic review by Teixeira et al. [13] reported consistent support for the positive relation between more autonomous forms of motivation to exercise, with a trend towards identified regulation predicting initial/short-term adoption more strongly than intrinsic motivation, and intrinsic motivation being more predictive of long-term exercise adherence. Indeed, a qualitative study assessing the motives and barriers to physical activity in older adults of different socioeconomic status highlighted the importance of integrated and identified regulations to promote physical activity engagement [14].

Zumba^®^ was reported in the top six favorite physical activities of midlife and older women [15]. Zumba Gold^®^ is recommended for the over-50s because it has lower-impact movements. In contrast to the growing worldwide participation in Zumba Gold^®^, there is limited evidence about its specific effects in postmenopausal women, with only two research groups investigating this area [4,16,17,18,19]. Their findings showed some significant benefits, including better balance, functional mobility, walking speed, lower limb muscular strength, quality of life, and mood after the intervention, compared to control groups who did not take part in Zumba Gold^®^ programs [4,16,17,18,19]. However, none of these studies focused on women from low SES, and they have not explored the effects of Zumba Gold^®^ on body composition, cardiovascular fitness, and mental health, which are known to change after the menopause [1,2,5].

Other studies on middle-aged and older women reported that Zumba Gold^®^ interventions led to decreases in body fat percentage and systolic blood pressure, improved flexibility and coordination, decreased anxiety and reduced barriers to physical activity, and improved intrinsic motivation to exercise, with contrasting results on cardiovascular fitness [4,20,21,22]. However, these results were not specific to postmenopausal women, with some studies including men [20], pre- and postmenopausal women together [22], or the inclusion of some health conditions, such as diabetes [21].

The studies described above show that exercise interventions based on Zumba Gold^®^ could be beneficial for various aspects of older women’s lives, and could be applied to women from low SES. However, most of these studies have focused on one particular aspect of women’s health (physical or psychological), highlighting the need for multidisciplinary research, and in particular, a greater focus on psychological outcomes and qualitative analyses. Therefore, the objective of this study was to investigate a Zumba Gold^®^ community-based intervention on the physical and psychological effects in postmenopausal women from low SES, with a multidisciplinary mixed-design approach.

## 2. Materials and Methods

### 2.1. Participants

Fifty postmenopausal women volunteered to take part in this study. They were recruited from six areas of low SES (deciles 1, 2, or 3) based on the 2019 Index of Multiple Deprivation (IMD) report for Great Britain. An a priori power calculation (G*Power 3.1, [23]), with an effect size of 0.5, an alpha level of 0.05, and a power = 0.8, suggested a sample size of 34. Postmenopausal was defined as an entire year without menstrual bleeding (www.nhs.uk, accessed on 5 October 2024). Inclusion criteria were not meeting the guidelines for physical activity for this age group (at least 150 min of weekly moderate intensity activity or 75 min of vigorous intensity, www.nhs.uk, accessed on 5 October 2024) as assessed with a physical activity diary. Exclusion criteria were any pulmonary, cardiovascular, or musculoskeletal contraindications to exercise and any severe impairment in cognitive function resulting in the inability to follow instructions (MiniMental State Examination (MMSE) score lower than 20 [24]). Participants were randomly allocated (50%/50%) to a Zumba Gold^®^ (ZG) group or a control (C) group by an online research randomizer (https://randomizer.org, accessed on 5 October 2024).

Two participants from the ZG group and five participants from the C group dropped out during the intervention, and hence the characteristics of the participants who finished the intervention were: n = 23; 68.7 ± 6.9 years old; 159.8 ± 4.9 cm; 65.7 ± 10.6 kg, 35.3 ± 10.3% of body fat for the ZG group and n = 20; 69.4 ± 7.4 years old; 157.1 ± 7.2 cm; 63.1 ± 11.9 kg, 35.9 ± 6.6% of body fat for the C group. Each participant was informed in detail about the testing procedures and possible risks of the study before signing an informed consent. The study was approved by the local University ethics committee in accordance with the principles set forth in the Helsinki declaration (University Research Ethics Committee, approval number 201495, 13 January 2022).

### 2.2. Procedures

This study was a randomized controlled trial, reported according to CONSORT, and followed a repeated measures design, with pre- and post-testing sessions, separated by a 12-week intervention period.

#### 2.2.1. Pre-And Post-Testing Sessions: Physical Parameters

Height (m) was measured with a portable scale (Seca, Marsden, UK), while body mass (kg), body fat (BF, %), visceral fat, and muscle mass (kg) were measured by bioelectrical impedance using the Tanita BC 418MA Segmental Body Composition Analyze (Tanita Corporation, Tokyo, Japan).

Baseline brachial blood pressure was recorded with participants resting for at least 5 min. They were seated, with their bare upper left arm supported at the level of the heart, and an automatic blood pressure device was used (Omron M2 Basic, Kyoto, Japan). The means of two measurements of systolic (SBP, mmHg) and diastolic (DBP, mmHg) blood pressure, taken 5 min apart, were calculated [25].

The 6-min walk test (6MWT) was used to measure participants’ cardiorespiratory fitness. It consisted of walking as fast as possible between two cones placed 10 m apart for six minutes. The total distance covered was measured. This test has shown good validity and reliability (test–retest correlation coefficient of r = 0.88) in older adults [26].

Lower body muscular strength was assessed by the 30 sit-to-stand (STS) test. It consists of sitting on a high chair, 17 inches in height, with arms crossed on the chest, standing up fully, and returning to a fully seated position as many times as possible for 30 s (s). This test is reliable (test–retest correlation coefficient of r = 0.89 in older women) and characterized by good criterion validity [26].

Balance was assessed by the Berg Balance Scale and the Y-Balance Test. The Berg Balance Scale is a test used to assess functional balance. It evaluates both dynamic and static balance through 14 tasks [27] and has very good reliability (test–retest correlation coefficient of r = 0.97 [28]). The lower Quarter Y-Balance Test was used to measure dynamic balance. It consists of maintaining balance while standing on one leg, with the contralateral leg reaching in three different directions (anterior, posteromedial, and posterolateral). The test was repeated twice, and the distance reached for each of the three directions was measured, and the sum was calculated as the main performance outcome. Freund et al. [29] reported good reliability for the mean reach distances in each direction (intraclass correlation coefficients ranging from 0.94 to 0.99).

#### 2.2.2. Pre-And Post-Testing Sessions: Psychological Parameters and Semi-Structured Interviews

Validated questionnaires were used to assess different aspects of quality of life and psychological well-being. These took between 30 and 45 min to complete.

-The Short Form Health Survey (SF-12) questionnaire was used to evaluate perceived physical and mental health. Two composite scores of physical and mental health (PCS and MCS) were calculated with a possible range from 0 to 100, with a higher score indicating better health. The SF-12 has demonstrated good test–retest correlations (0.76–0.89) and a sub-scale reliability of >0.70 [30], with responsiveness to change [31].-The Multidimensional Fatigue Index (MFI-20) questionnaire was used to assess five aspects of fatigue: general fatigue, physical fatigue, reduced activity, mental fatigue, and reduced motivation [32]. Each item was scored from 1 to 5, with four items per subscale. It is characterized by very good reliability (Intraclass correlation coefficients > 0.80 [33] and good convergent validity (correlation coefficients with a visual analogue scale measuring fatigue ranging between 0.22 and 0.78 [32]).-The 9-item Self-Efficacy for Exercise (SEE) questionnaire was used to assess an individual’s confidence in their ability to engage in exercise. Each item is rated from 0 (Not confident) to 10 (very confident) with a mean score (0–10) indicating the self-efficacy to exercise despite barriers (e.g., poor weather). The total score was obtained by summing the scores for the 9 items. The SEE has demonstrated very good internal consistency (0.89 to 0.92) and high test–retest reliability (0.80–0.92) [34].-The 19-item Behavioral Regulations in Exercise Questionnaire (BREQ-2) was used to measure motivation to exercise [35]. It is divided into five subscales: Amotivation (lack or absence of motivation), External regulation (driven by external rewards or pressures), Introjected regulation (engage due to internal pressures, such as guilt, shame, or the need to maintain self-worth), Identified regulation (engage because recognize and accept its personal value or importance) and Intrinsic motivation (engage for its own sake, out of genuine interest, enjoyment, or personal satisfaction). A relative autonomy index (RAI), representing the degree to which participants feel self-determined, was also calculated by applying a weighting to each subscale and then summing the weighted scores. The BREQ-2 was shown to be a valid and reliable tool with strong subscale internal consistency (Cronbach’s alpha ranging from 0.60 to 0.88) [36].-Anxiety and depression were recorded using the Hospital Anxiety and Depression Scale (HADS [37]), consisting of seven anxiety-focused items and seven depression-focused items, each scored as a Likert (1–5) Scale. It has good construct validity and internal consistency reliability (Cronbach’s alpha values of 0.890 for the anxiety scale and 0.856 for the depression scale) [38].-The Social Connectedness Scale [39] was used as a measure of positive well-being. It includes eight items, each scored on a Likert (1–6) Scale. It has shown very good internal consistency (Cronbach’s alpha of 0.91) and very good test–retest reliability (correlation coefficient of 0.96).

##### Semi-Structured Interviews

Following the intervention, fifteen participants in the Zumba Gold^®^ group took part in a semi-structured interview to gain feedback on the intervention and potential physical and psychological impact of the exercise program. Questions were asked about enjoyment of the classes, perceived physical and psychological benefits, sense of community/belonging, new friendships, as well as barriers and facilitators to attending, injuries or pain resulting from the classes, and their willingness to continue after the study.

#### 2.2.3. Zumba Gold^®^ Intervention

The Zumba Gold^®^ intervention required participants to take part in two weekly Zumba Gold^®^ classes in their community for a period of 12 weeks. They were also advised not to change their usual physical activity outside the program during the intervention period, and to complete a simple physical activity diary (type and duration of exercise weekly). Prior to the study, the research team found several suitable (in the main community or neighborhood center, i.e., walking distance from participants’ homes) and affordable classes in the six areas where participants were recruited from. The cost of classes varied from £1 to £3 per class, as part of an initiative from these neighborhoods to promote physical activity, independently from the study. A total of ten classes, led by five different qualified instructors were used. This was to capture the variety of practices between instructors and avoid the bias of a single class approach. Each participant had their heart rate (HR) measured during four different classes during the 12 weeks to get a more representative idea of their physiological responses. Each session lasted 60-min and included one or two warm-up and cool-down songs, and the main part of the session was structured around steps from the following six dance styles commonly used in Zumba: merengue, cumbia, reggaeton, salsa, belly dancing, and pop. Short breaks (<30 s) were given between most songs, and participants were encouraged to drink water. For four sessions, participants were required to arrive at the class early and fitted with an HR monitor (Polar H7, Polar, Kempele, Finland). Participants’ HR were recorded at 1-s intervals during Zumba Gold^®^ classes.

From the recordings, the average HR (HR_mean_) was calculated, using the average HR values from the start of the warm-up to the start of the cool-down. These values were expressed in absolute (beat min^−1^) and relative (% of maximal HR (HR_max_)), according to the Tanaka equation [40]. Finally, the time spent in the following HR zones was calculated [41]:-Zone 1 (very light-to-light): HR < 64% of HR_max_;-Zone 2 (moderate) 64% of HR_max_ ≤ HR ≤ 76% of HRmax;-Zone 3 (vigorous to maximal): 76% of HR_max_ ≤ HR ≤ 95% of HR_max_;-Zone 4 (maximal): HR < 95% of HR_max_.

During the 12-week intervention period, the control group carried on their usual daily activities (including physical activity or not) without taking part in Zumba Gold^®^ classes. Similarly to the Zumba Gold^®^ group, they were required to report their weekly physical activity type and duration in a diary.

#### 2.2.4. Statistical Analyses

Data was presented as mean ± standard deviation, with 95% confidence intervals (95% CI). Normality was checked by the Shapiro–Wilk test, and all data were normally distributed (p ranging from 0.098 to 0.978). Differences between groups in the weekly duration of physical activity during the intervention and differences between groups in all outcome variables at baseline were assessed by a T-test for independent samples, showing significant differences between groups for the anxiety element of the HADS, the external regulation component of the BREQ-2, and the Y-balance score on the left leg (*p* < 0.05). Consequently, an analysis of covariance (ANCOVA) was performed on all outcome variables to assess the effects of group (ZG vs. C) on the post-intervention data, with the baseline data as a covariate. Effect sizes were calculated as Partial Eta Squared (ƞp^2^) and interpreted as no effect (0–0.05), minimum effect (0.05–0.26), and strong effect (0.26–0.64) [42].

Thematic analysis was used for the qualitative data. Braun and Clarke [43]’s thematic analysis method elicited six steps in analyzing the data: (1) becoming familiar with the data, (2) generating codes, (3) generating themes, (4) reviewing themes, (5) defining and naming themes, and (6) report write-up. Transcripts were read numerous times by 2 authors (LS and AB) to facilitate data immersion, increase content familiarity, and ensure accuracy. Nvivo was used to synthesize the data, generating codes, sub-themes, and themes. One of the authors conducted the initial coding (LS), which was then iteratively reviewed and discussed with a second author (AB). Confirmability [44] was enhanced by the author’s (LS) reflexive self-awareness. The author recognized any researcher bias [45] by considering any perspectives that were evident and how these may have impacted data collection and analysis [44]. Reflexivity and sincerity were enhanced through a second author (AB) acting as a critical friend to encourage reflection and exploration of alternative interpretations [44].

## 3. Results

Seven out of our 50 initial participants (14%) dropped out of the study for various reasons (n = 4 personal reasons, n = 2 medical reasons, n = 1 an unknown reason). Overall, compliance with the Zumba Gold^®^ intervention was 65 ± 25%. The type of physical activity undertaken by both groups (outside the intervention for the ZG group) included walking, cycling, yoga, Pilates, and swimming in the C group, and walking, yoga, Pilates, and swimming in the ZG group. The statistical analysis did not show any significant difference between groups in the mean weekly duration of physical activity, outside the exercise program (102 ± 48 min vs. 88 ± 56 min, respectively, for the C and ZG groups, *p* > 0.05).

The mean HR during the Zumba Gold^®^ classes was 70.2 ± 6.0% of HR_max_, with 25.5 ± 24.7%, 48.0 ± 19.3%, 25.6 ± 20.1%, and 0.9 ± 3.1% of the session spent in HR zones 1, 2, 3, and 4, respectively.

The effects of the exercise intervention on anthropometric, physiological, and physical performance variables are presented in Table 1. The ANCOVA showed a significant effect of group on muscle mass (F(1) = 4.297, *p* = 0.045, ƞp^2^: 0.102), STS (F(1) = 6.595, *p* = 0.014, ƞp^2^: 0151) and 6MWT (F(1) = 7.659, *p* = 0.009, ƞp^2^: 0168), with greater muscle mass and STS repetitions and lower 6MWT time in the ZG group compared to the C group (Table 1, Figure 1 and Figure 2). No significant group effect was observed on any other anthropometric, physiological, or physical performance variables (*p* > 0.05, Table 1).

The effects of the exercise intervention on psychological variables are presented in Table 2 and Table 3. The ANCOVA showed a significant effect of group on the MCS component of the SF-12 (F(1) = 11.367, *p* = 0.002, ƞp^2^: 0226), with significantly greater values in the ZG compared to the C group (Table 2). Significant group effects were also observed for the general fatigue (F(1) = 9.860, *p* = 0.003, ƞp^2^: 0198), physical fatigue (F(1) = 8.814, *p* = 0.005, ƞp^2^: 0181), and mental fatigue (F(1) = 8.007, *p* = 0.007, ƞp^2^: 0167) components of the MFI-20. Specifically, the scores on these components were significantly lower in the ZG group compared to the C group (Table 2). Furthermore, the ANCOVA showed a significant group effect on the Social Connectedness Scale (F(1) = 10.718, *p* = 0.002, ƞp^2^: 0216), with a significantly greater score in the ZG group compared to the C group (Table 2). Finally, results on the BREQ-2 showed a significant group effect on the amotivation component only (F(1) = 9.846, *p* = 0.003, ƞp^2^: 0202, Table 3), with a significantly lower value in the ZG compared to the C group (Table 3). No significant group effect was observed on any other psychological variables (*p* > 0.05, Table 2 and Table 3).

Four main themes were identified from the semi-structured interviews (Table 4): belonging, non-judgmental place, psychological motivational factors, and mind–body connection. The analysis demonstrated that the ZG exercise program had a positive effect on all four main themes. Two to four sub-themes were generated under these main themes, and again, the exercise intervention had positive effects on these. Some quotes from participants are shown in Table 4.

## 4. Discussion

The main results of the present study showed that a 12-week exercise program based on Zumba Gold^®^ led to significant improvements in lower limb strength (STS test) and cardiorespiratory endurance (6MWT) as well as psychological outcomes, including a significant better mental health score on the SF-12 questionnaire, significantly lower general, physical and mental fatigue components of the MFI-20, a significantly greater social connectedness score and a significantly lower amotivation score on the BREQ-2, compared to the control group. Finally, the qualitative analysis of the participant interviews revealed that our intervention had positive effects on four main themes: belonging, non-judgmental place, psychological motivational factors, and mind–body connection. It should be noted, however, that the changes observed in the present study were all associated with minimum effect sizes.

The significantly greater performance in the STS test observed in the present study is in line with previous findings. Indeed, Ben Waer et al. [18] and Lahiani et al. [19] found significant improvements in STS performance following 12-week exercise programs based on Zumba Gold^®^ in middle-aged and postmenopausal women. The slightly greater improvements observed in the studies of Ben Waer et al. [18] and Lahiani et al. [19] compared to the present study (+13.6% and +17.3%, vs. +9.0%, respectively, in the Ben Waer et al. [18], Lahiani et al. [19], and our study) could also be explained by the greater number of weekly sessions (three) performed in these studies compared to ours (two). The elements specific to Zumba Gold^®^ that could explain the better STS performance following interventions are the inclusion of body weight resistance training elements, such as squats or lunges. Indeed, bodyweight resistance training increases muscle mass and strength performance [46,47] Within this context, we observed a significantly greater post-intervention muscle mass in our ZG group compared to the C group. Increasing muscle mass is particularly important in postmenopausal women because sarcopenia starts to occur around the menopause, and could lead to frailty and falls later in life [1,48]. Factors other than muscle mass could also explain the better STS performance in the ZG group, such as the neuromuscular adaptations usually observed with strength training [49], or potentially greater cardiorespiratory fitness that could influence the feeling of fatigue during the repetitions.

To our knowledge, our study is the first to show that a Zumba Gold^®^ exercise program can significantly improve cardiorespiratory fitness, assessed by the 6MWT, specifically in healthy sedentary postmenopausal women. Other studies on slightly younger women with health conditions or healthy mixed gender groups of the same age as our participants showed contrasting results. Indeed, significant improvements in maximal oxygen uptake and submaximal aerobic performance were shown following exercise programs lasting between 12 and 16 weeks and based on Zumba Gold^®^ in overweight middle-aged women with diabetes and healthy men and women aged 65 to 75 years old [20,21]. However, Rossmeissl et al. [22] did not find any significant effects of a 12-week Zumba Beat (similar to Zumba Gold^®^) exercise program on maximal oxygen uptake in overweight women aged 45 to 65 years old. These contrasting results could be due to the poor adherence to the exercise program reported by Rossmeissl in their discussion of the results (no numerical data for compliance reported). In contrast, the present study showed good average compliance with the training program (65 ± 25%), despite a relatively large standard deviation [50]. This good adherence might be related to the thematic findings linked to belonging, non-judgment, enjoyment, and well-being, creating a virtuous cycle of engagement and appreciating the immediate and sustained benefits for some of the women. In contrast, our relatively high variability in adherence could be explained by socioeconomic factors (financial constraints, integration of exercise in working and daily life), social factors (lack of support network), and psychological factors that affect the perceived self-efficacy of women from low SES (negative body image, fear of judgment) [9,10,11]. Training intensity has also been cited in the literature as an important determinant of the benefits of training on cardiorespiratory fitness [51]. Within this context, our average HR was 70.2% of HR_max,_ and our participants spent 74.5% of class time in HR zone 2 (moderate intensity) or above, which corresponds to 40.8 min. These results are in line with the recommendations of the American College of Sports Medicine (ACSM) to spend 30-min daily performing moderate-intensity exercise in order to improve/maintain cardiorespiratory endurance [41]. The women’s positive feedback suggests that Zumba Gold^®^ is a feasible approach for supporting physical and psychological well-being post-menopause for women from lower SES settings.

Our results showed no significant effect of our intervention on body fat, suggesting that 12 weeks of Zumba Gold^®^ performed twice weekly is not enough to induce weight loss. Indeed, while heart rate in training zone 1 and 2 is optimal to improve fat oxidation, the limited duration, frequency, and training volume, and other factors such as our participants’ age and their diet may have influenced this non-significant result. For example, Barbalho et al. [52] compared the effects of a 12-week resistance training program at low vs. high volume in healthy older women and showed greater decreases in body weight and waist circumference in the high-volume group. In addition, age is a well-known factor to impede weight loss in middle-aged and older women, mainly due to the effects of the menopause on muscle mass and metabolism [3,4]. Finally, our intervention did not include any control or modification of our participants’ diet, which could further explain the lack of significant changes in body fat in the ZG group. Indeed, many studies reported the benefits of calorie restriction in addition to physical exercise on weight loss compared to exercise alone [53]. To our knowledge, the only study that reported a significant decrease in fat mass after 16 weeks of Zumba Gold^®^ performed three times weekly focused on overweight and obese middle-aged women [21]. While it is common for menopausal women to want to lose weight, due to metabolic changes and fat distribution associated with lower estrogen levels [2], our participants were not overweight, and hence this was not a specific objective of the present study. It should be noted, however, that the ZG increased their body fat between baseline and post-intervention, which could lead to greater metabolic risk, although this change was non-significant. Similarly, our results did not show any significant benefits of our intervention on static and dynamic balance. This is surprising given that the complex sequence of steps involved in the dance routines of Zumba Gold^®^ has been shown to improve balance through better core strength and proprioception, in particular [54]. Our results are in contrast with previous studies that observed significant improvements in balance after Zumba Gold^®^ interventions of similar duration to ours [16,17,18,19]. The lack of significant results in the present study could be explained by the tests used to measure balance. Indeed, while the Berg test is widely used to measure balance and assess the risk of falls in older individuals, it is also characterized by a ceiling effect in people with higher levels of physical performance, because the task items may not be challenging enough for them [55]. It is possible that we would observe a significant difference between groups if we had used a more sensitive measure, such as a posturograph, for example. Within this context, our participants achieved close to maximal performance at baseline (54.8 ± 1.4 and 54.5 ± 1.5, for a maximal score of 56). We also used the Y-balance test which may not allow people with some musculoskeletal issues or a lack of strength to reach a certain distance, despite good balance [56]. In contrast, studies that showed a significant effect of Zumba Gold^®^ on balance used force platforms to measure center of pressure sways in stable and unstable conditions [16,17,18,19], providing a more objective measure relying less on other factors, such as lower limb strength. Nonetheless, interviews identified some subjective improvements in perceived fitness and energy levels.

The results of the present study showed significant improvements in psychological outcomes following our intervention. We observed a significant increase in the perceived mental health component of the SF-12 questionnaire, and significant decreases in general, physical, and mental fatigue elements of the MFI-20 questionnaire. This was supported by the qualitative data with the indication of improved mood and energy levels: “feeling good factor lasts into the evening”. Similar benefits of a Zumba Gold^®^ intervention were reported in postmenopausal women after 12 weeks, specifically a significant improvement in the perceived mental health component of the longer version of the SF-12 (SF-36, [19]). Other studies also reported significant improvements in some components of the SF-36 in women aged 55–80 years old [4] and significant decreases in the physical and mental fatigue elements of the MFI-20 [20] in older men and women following exercise interventions based on Zumba Gold^®^. These authors suggested that these improvements reflect a better Quality of Life following Zumba Gold^®^ and attributed these benefits to the specificity of this type of workout [4,19,20]. Specifically, the variety of Latin dance choreographies, dynamic music, and fun atmosphere in Zumba Gold^®^ classes are thought to trigger pleasant emotions [57]. In the present study, our participants mentioned that “it (Zumba Gold^®^) makes you happy, right? Exercise is like an antidepressant”. In addition, Zumba Gold^®^ includes relatively vigorous physical exercise (25.6% of class time (about 15-min) spent in the vigorous to maximal HR zone in the present study, which is in line with the ACSM recommendations to spend 75-min weekly performing vigorous-intensity exercise if this was repeated daily [41]). This is due to its relatively fast pace, involvement of major muscle groups of the upper limb, lower limb, and trunk, and frequent changes of direction. Vigorous exercise is known to increase endorphin levels, even when estrogen concentrations are decreasing [58]. This is achieved by an activation of the body’s stress response, mainly through the hypothalamic-pituitary-adrenal axis, leading to the release of endorphins [58]. Despite our findings of significant improvements in perceived mental health and decreased mental fatigue, no significant changes in anxiety or depression were shown in the present study. It is in contrast with the significantly lower anxiety scores of the HADS observed by Kasim et al. [20] following 12 weeks of Zumba Gold^®^. However, these authors also mentioned that an absence of significant change in the HADS outcomes (which they observed for the depression sub-component in their study) could be linked to a low baseline score. This could explain our results, as our scores for both components of the HADS are below 7.0, which is considered a lack of significant anxiety and depression (>8 indicates mild) [37].

Our findings showed a significant improvement in The Social Connectedness Questionnaire score following intervention in the ZG group only. The fun and social benefits of Zumba^®^ and Zumba Gold^®^ were highlighted by many authors in a range of populations [59,60,61,62]. Delextrat et al. [59] interviewed older adults with Parkinson’s after undertaking a six-week Zumba Gold^®^ intervention, and participants mentioned that Zumba Gold^®^ was “fun and novel”, that they “enjoyed meeting new people”. This is in line with the theme of “belonging” identified in our qualitative analysis, with some of our participants mentioning that the class was “a new community, like a therapy place” or “you enjoy being with them” (Table 4). In addition, a systematic review on the effects of Zumba reported significant social benefits on Quality of Life, and specifically a significant improvement in the Social Functioning domain of the SF-36 questionnaire [62]. Social connectedness is particularly important in older adults with low SES who are reported to experience higher levels of loneliness [63,64]. Several recent interventions in the community are being developed to improve these aspects, and significant benefits of some (not based on exercise) have been shown on well-being and quality of life of older adults from low SES. These effects were mediated by social connectedness [63]. A neighborhood-based intervention based on physical activity is currently being developed to promote healthy ageing in low SES older adults, but results have not been published yet [64].

In relation to our findings of better social connectedness, we also observed a significantly lower level of amotivation in the ZG group. As amotivation reflects apathy and no intention to exercise, a low level of this type of motivation is perceived to be a positive outcome. Linking back to the SDT and the three basic needs of autonomy, competence, and relatedness, the intervention certainly appears to have increased the relatedness aspect, i.e., increased social connectedness scores. This was further supported by the qualitative theme of “belonging” and “relationships” with quotes such as “We share interests and go for coffee, whereas I’ve never met them outside the gym”. Surprisingly, this increased connectedness did not increase other aspects such as extrinsic motivation, intrinsic motivation, or a change in the relative autonomy index. Indeed, increased motivation to exercise following Zumba or Zumba Gold^®^ intervention has been reported previously [19,55]. Krishnan et al. [21] reported an improvement in intrinsic motivation to exercise in overweight middle-aged women. In accordance with Krishnan et al., Delextrat et al. [58] highlighted greater autonomy (+8.0%) and purpose in life (+4.4%) following 8 weeks of Zumba in young women. The differences between our findings and those of others could be due to the different populations focused on in these papers, and suggest that older female adults from low SES might not feel the competence and autonomy aspect during a 12 week intervention especially if they have not previously participated in physical activity classes due to low SES specific barriers such as cost and access [10,11]. Additionally, the adherence rates may have also limited the potential for basic needs development. Indeed, a study by Bukan et al. [65] compared focus group responses from high and low SES and reported that low SES individuals seemed to be especially motivated to change their lifestyle when they experienced health complaints, but were rather hesitant to change their lifestyle for preventive purposes. This may reflect the lower levels of amotivation in the current ZG group, but the lack of increase in the other, more adaptive forms of motivation. It should also be noted that the ZG group had relatively low levels of amotivation at baseline. Nevertheless, Bukan et al. [65] also reported that individuals in the low SES group preferred physical activities where groups consisted of individuals of the same age, gender, and physical condition. These characteristics were certainly evident in the current exercise program.

There are a few limitations in the present study. First, our participants had to pay for Zumba Gold^®,^ and we did not measure if this was a barrier to attendance. The large standard deviation in the adherence to classes may be a reflection of this, but we did not ask participants. It is possible that the benefits of our intervention could have been greater if adherence to classes had been higher. Furthermore, our measure of balance had some limitations, with the Berg test showing a ceiling effect, which could have biased the results towards a lack of significant effect. Nevertheless, we controlled for this potential bias through the use of ANCOVAs controlling for baseline levels. We also used a variety of classes, instructors, and settings, which introduces some variability in our intervention and we only measured the physical activity performed by both groups outside the study by diaries. The use of accelerometers would have provided a more valid assessment. Another potential limitation is that we compared our ZG group to individuals who did not take part in any structured exercise program. It could be interesting to use a comparison group involved in another type of exercise, such as high-intensity interval training (HIIT), for example. Finally, we did not assess all the menopausal symptoms, and it would be interesting in future research to focus on somatic (i.e., hot flashes, joint pain) or urogenital symptoms, for example.

## 5. Conclusions

In conclusion, our findings showed that a 12-week exercise program based on Zumba Gold^®^ led to significant improvements in physical (lower limb strength, cardiorespiratory endurance, muscle mass) and psychological variables (mental health, fatigue, social connectedness, and motivation to exercise) in women from low SES. We also observed positive effects on four main themes, including belonging, non-judgmental place, psychological motivational factors, and mind–body connection. However, no significant effects of the intervention were shown on blood pressure, body fat, balance, anxiety, depression, and self-efficacy to exercise. These findings suggest that Zumba Gold^®^ is an appropriate exercise for this population, and the use of HR can be helpful to monitor exercise intensity and maximize sessions. Further studies should be undertaken to look into these aspects and compare the effects of a similar intervention in women from various SES. It could be interesting, in particular, to compare Zumba Gold^®^ to other types of planned exercise (HIIT, concurrent training, etc). In addition, the effects of Zumba Gold^®^ on other aspects that are important around the menopause, such as weight management, sleep, and cognition, should be investigated.

## Figures and Tables

**Figure 1 healthcare-13-02250-f001:**
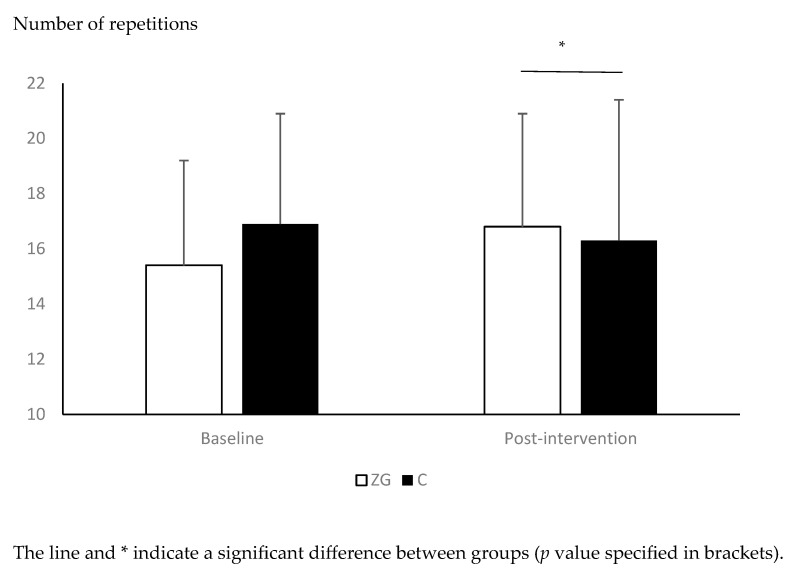
Changes in sit-to-stand (STS) performance between pre- and post-intervention in the Zumba Gold^®^ (ZG) and control (C) groups.

**Figure 2 healthcare-13-02250-f002:**
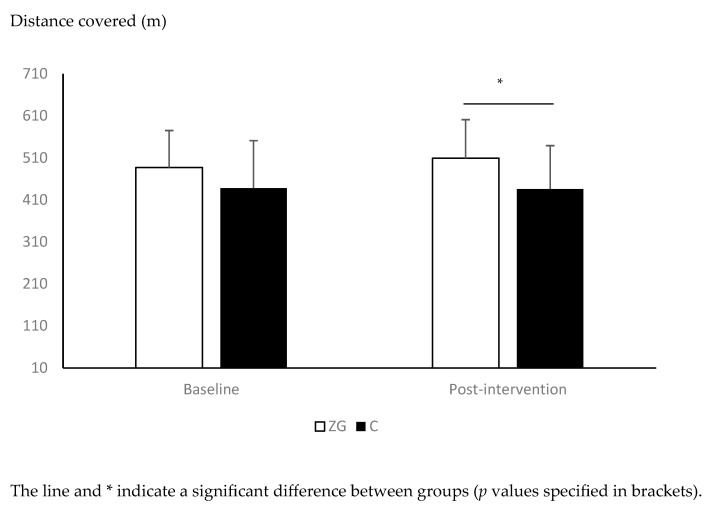
Changes in six-minute walk test (6MWT) performance between pre- and post-intervention in the Zumba Gold^®^ (ZG) and control (C) groups.

**Table 1 healthcare-13-02250-t001:** Changes in anthropometric, physiological, and physical performance variables between pre- and post-intervention in the Zumba Gold^®^ (ZG) and control (C) groups (SBP: systolic blood pressure, DBP: diastolic blood pressure, STS: sit-to-stand test, 6MWT: 6-min walk test).

Variable	Group	Pre	Post	95% CI for the Difference Between Groups (SD)
Resting SBP (mmHg)	ZG	136 ± 20	127 ± 30	−11 to 15 (42)
	C	138 ± 19	138 ± 20	
Resting DBP (mmHg)	ZG	82 ± 18	80 ± 12	−9 to 8 (27)
	C	80 ± 9	78 ± 8	
Body mass (kg)	ZG	65.7 ± 10.7	65.0 ± 10.9	−1.7 to 0.6 (3.8)
	C	63.5 ± 12.5	63.3 ± 12.6	
Body fat (%)	ZG	35.3 ± 10.3	36.2 ± 8.4	−1.4 to 2.6 (6.5)
	C	35.4 ± 6.8	35.7 ± 7.2	
Visceral Fat Index	ZG	8.7 ± 3.5	8.6 ± 3.2	−0.6 to 0.9 (2.3)
	C	9.1 ± 2.8	8.8 ± 2.8	
Muscle mass (kg)	ZG	26.3 ± 3.8	26.9 ± 3.3	0.1 to 2.6 (4.2)
	C	26.2 ± 4.1	25.6 ± 4.0 *	
Y-balance right	ZG	81.0 ± 14.5	79.5 ± 16.1	−9.3 to 4.5 (22.3)
	C	72.7 ± 17.1	74.8 ± 17.9	
Y-balance Left	ZG	82.4 ± 14.2	79.4 ± 16.7	−10.3 to 5.5 (25.5)
	C	72.9 ± 19.3	74.7 ± 17.4	
Berg balance	ZG	54.8 ± 1.4	55.2 ± 0.9	−0.1 to 1.4 (2.3)
	C	54.5 ± 1.5	54.3 ± 1.9	

*: significant difference between groups, *p* < 0.05.

**Table 2 healthcare-13-02250-t002:** Changes in Short Form Health Survey (SF-12 PCS: physical component, MCS: mental component), Multidimensional Fatigue Index (MFI-20), self-efficacy for exercise (SEE), and social connectedness variables between pre- and post-intervention in the Zumba Gold^®^ (ZG) and control (C) groups.

Variable	Group	Pre	Post	95% CI for the Difference Between Groups (SD)
SF-12 PCS	ZG	48.6 ± 8.2	50.4 ± 9.5	−5.8 to 3.5 (15.0)
	C	47.7 ± 8.7	51.9 ± 6.9	
SF-12 MCS	ZG	49.2 ± 9.1	56.9 ± 5.4 **	3.2 to 12.8 (15.5)
	C	50.5 ± 7.6	49.2 ± 9.4	
MFI-20 General fatigue	ZG	9.7 ± 4.0	8.5 ± 3.3 **	−3.4 to −0.7 (4.3)
	C	9.7 ± 2.0	10.5 ± 2.5	
MFI-20 Physical fatigue	ZG	9.2 ± 3.6	7.9 ± 3.1 **	−3.3 to −0.6 (4.4)
	C	9.3 ± 3.1	10.0 ± 2.8	
MFI-20 Reduced activity	ZG	7.9 ± 3.2	7.5 ± 3.8	−2.7 to 0.6 (5.3)
	C	9.0 ± 2.7	9.3 ± 2.5	
MFI-20 mental fatigue	ZG	8.0 ± 3.0	6.9 ± 3.4 **	−3.0 to −0.5 (4.1)
	C	7.2 ± 2.6	7.9 ± 3.0	
MFI-20 reduced motivation	ZG	8.0 ± 3.2	7.9 ± 3.9	−1.9 to 1.3 (5.1)
	C	9.0 ± 2.3	9.1 ± 3.6	
SEE	ZG	51.8 ± 20.2	53.6 ± 19.2	−8.7 to 8.9 (28.5)
	C	51.7 ± 17.3	53.5 ± 17.5	
Social connectedness	ZG	40.7 ± 9.9	44.7 ± 5.0 **	1.5 to 6.3 (7.8)
	C	41.3 ± 7.4	41.0 ± 5.0	

**: significant difference between groups, *p* < 0.01.

**Table 3 healthcare-13-02250-t003:** Changes in Behavioral Regulation in Exercise Questionnaire (BREQ-2) and Hospital Anxiety and Depression Scale (HADS-A and HADS-D) variables between pre- and post-intervention in the Zumba Gold^®^ (ZG) and control (C) groups.

Variable	Group	Pre	Post	95% CI for the Difference Between Groups (SD)
BREQ-2 Amotivation	ZG	0.03 ± 0.11	0.01 ± 0.05 **	−0.60 to −0.13 (0.76)
	C	0.37 ± 0.66	0.49 ± 0.69	
BREQ-2 External Regulation	ZG	0.13 ± 0.29	0.39 ± 0.61	−0.07 to 0.56 (1.02)
	C	0.38 ± 0.50	0.26 ± 0.39	
BREQ-2 Introjected Regulation	ZG	1.27 ± 0.64	1.06 ± 0.95	−0.50 to 0.22 (1.17)
	C	1.14 ± 0.80	1.13 ± 0.55	
BREQ-2 Identified Regulation	ZG	3.43 ± 0.45	3.39 ± 0.56	−0.31 to 0.41 (1.16)
	C	3.14 ± 0.69	3.14 ± 0.78	
BREQ-2 Intrinsic Regulation	ZG	3.64 ± 0.48	3.61 ± 0.59	−0.28 to 0.46 (1.20)
	C	3.33 ± 0.95	3.30 ± 0.94	
BREQ-2 RAI	ZG	16.0 ± 3.1	16.3 ± 2.9	−1.0 to 3.2 (6.7)
	C	13.5 ± 4.9	12.8 ± 5.5	
HADS-A	ZG	4.74 ± 2.88	3.91 ± 4.00	−1.94 to 1.41 (5.42)
	C	6.95 ± 2.95	6.60 ± 3.28	
HADS-D	ZG	4.26 ± 3.12	3.65 ± 3.37	−1.42 to 1.65 (4.97)
	C	4.80 ± 2.53	4.05 ± 2.33	

**: significant difference between groups, *p* < 0.01.

**Table 4 healthcare-13-02250-t004:** Themes, subthemes, and quotes for the effect of physical and psychological well-being experiences in Zumba Gold^®^ classes.

Themes	Sub Themes	Quotes
Belonging	Extension of community	“You enjoy being with them”.
	Friendships and acquaintances	“That’s why I’ve started going because I know people who are going, it’s local people”.
	Shared interest	“You’re interested in things; you’re not somebody who goes home and watches football”.
	Therapeutic space	“A new community like a therapy space”.
Non judgmental space	Positive body image	“Don’t worry, you know if you make a mistake, you just do a hop and carry on”.
	Validation	“They (sic) I need to watch her all the time”.
	Mistakes are allowed	“You do the room movements, and you bump into someone, but we carry on”.
Psychological motivational factors	Music natural mood enhancer	“We’re just really happy, music, it’s happy music”.
	Relationships	“We share interests and go for coffee, whereas I’ve never met them outside the gym”.
	Confident growth mindset	“Remember, the dance is each one, then you gain more confidence”.
	Happiness—feels good	“It makes you happy, right?” “Exercise is like an antidepressant”.
Mind–body Connection	Physical fitness	“I can get the physical wellness”.
	Cognitive coordination	“Cha, Cha 1.2.3, 1,2,3 it’s like dancing but learning less repetitive it’s not the same action”.
	Improved energy levels	“Feeling good factor lasts into the evening”.

## Data Availability

The data supporting this publication can be found in the data repository below: https://radar.brookes.ac.uk/radar/items/861ebf4a-0c97-4946-8b59-f8930e802666/1/, accessed on 14 July 2025.

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
