# Peer review of "Physical and Psychological Benefits of a 12-Week Zumba Gold® Exercise Intervention in Postmenopausal Sedentary Women from Low Socioeconomic Status"

_healthcare, 2025, doi:10.3390/healthcare13172250_

Round 1
Reviewer 1 Report
Comments and Suggestions for Authors The manuscript addresses an important topic by investigating both physical and psychological benefits of Zumba Gold® in postmenopausal women from low socioeconomic backgrounds. The study’s methodology is generally sound, and the mixed-methods approach is a clear strength. However, to meet the standards of a Q1 journal such as Biology, the manuscript requires significant revisions on several levels:— language (non-native phrasing, grammatical issues),
— structural clarity,
— logical interpretation of results,
— as well as refinement of scientific argumentation.
Language and Style Issues
Redundancy: Many phrases are repeated verbatim in the abstract and introduction (e.g., “community-based intervention in postmenopausal women from low SES”).
Grammatical errors:
“Zumba Gold had positive effects of four main themes…” → should read “on four main themes.”
“...the main body was based on steps…” → better phrased as “the main part of the session was structured around steps…”
Non-native expressions: e.g., “This test is characterised by...” → Replace with: “This test has shown...”Content and Logical Observations
1. Novelty and Objectives
- The novelty claim (“this is the first study…”) is overused and not sufficiently supported by citations.
- The contribution should be clearly articulated: low SES sample + psychological outcomes + qualitative depth.
-
Lack of clarity regarding the activities of the control group.
- Suggest adding whether physical activity was monitored (e.g., via diary or questionnaire).
-
Multiple instructors and settings are used (10 classes, 5 instructors), which may introduce variability.
- This should be acknowledged in the Limitations section.
-
The finding of increased external regulation is treated as a positive outcome. However, this is ambiguous without contextualisation.
- Recommend linking to Self-Determination Theory (Deci & Ryan) and reflecting critically on the motivational profile shift.
Statistical and Methodological Points
- The RAI (Relative Autonomy Index) is calculated but not discussed.
- Normality testing is stated, but not reported. Suggest summarizing assumptions compliance.
- Balance outcomes may be affected by ceiling effects in the Berg Balance Scale. Suggest using more sensitive tools or reflecting this as a limitation.
Tables and Figures
-
Tables 1–3 are dense and hard to interpret.
- Recommend adding at least one figure (e.g., STS or MCS scores over time) for visual clarity—this is common practice in MDPI journals.
References and Theoretical Framing
- Core literature on exercise motivation (e.g., Deci & Ryan 2000) is missing.
Summary of Recommended Actions
- Substantial language and grammar revision throughout the manuscript.
- Strengthen the framing in the Introduction and Discussion, especially regarding novelty.
- Provide a more nuanced interpretation of psychological outcomes (especially motivation).
- Include at least one visualisation of the main results.
- Expand the theoretical discussion by integrating key frameworks from exercise psychology.
Author Response
Reviewer 1
The manuscript addresses an important topic by investigating both physical and psychological benefits of Zumba Gold® in postmenopausal women from low socioeconomic backgrounds. The study’s methodology is generally sound, and the mixed-methods approach is a clear strength. However, to meet the standards of a Q1 journal such as Biology, the manuscript requires significant revisions on several levels:
— language (non-native phrasing, grammatical issues),
— structural clarity,
— logical interpretation of results,
— as well as refinement of scientific argumentation.
Answer: thanks for your comments, we have addressed all your comments, and highlighted to changes in yellow in the text We have replied to your specific comments after each point (in red).
Language and Style Issues
Redundancy: Many phrases are repeated verbatim in the abstract and introduction (e.g., “community-based intervention in postmenopausal women from low SES”).
Grammatical errors:
“Zumba Gold had positive effects of four main themes…” → should read “on four main themes.”
“...the main body was based on steps…” → better phrased as “the main part of the session was structured around steps…”
Non-native expressions: e.g., “This test is characterised by...” → Replace with: “This test has shown...”
Answer: The text has been proof-read by a native co-author and changes are highlighted in yellow in the text. For your first comment, we changed the wording of the part repeated in the abstract and introduction.
Content and Logical Observations
1. Novelty and Objectives
- The novelty claim (“this is the first study…”) is overused and not sufficiently supported by citations.
Answer: We have rewritten a large part of the introduction (lines 51-91) and added several references, which should make this clearer. We have also not relied on this type of expression (the is the first study) as much in the newer version too
- The novel contribution should be clearly articulated: low SES sample + psychological outcomes + qualitative depth.
Answer: we have modified the last paragraph to highlight this (lines 112-119): “ The studies described above showed that exercise interventions based on Zumba Gold® could be beneficial for various aspects of older women’s lives, and could be applied to women from low SES. However, most of these studies have focused on one particular aspect of women’s health (physical or psychological), highlighting the need for multidisciplinary research, and in particular a greater focus on psychological outcomes and qualitative analyses. Therefore the objective of this study is to investigate the physical and psychological effects of a Zumba Gold® community-based intervention in postmenopausal women from low SES, with a multidisciplinary mixed-design approach.”
- Control Group Description
- Lack of clarity regarding the activities of the control group.
- Suggest adding whether physical activity was monitored (e.g., via diary or questionnaire).
Answer: Thanks for your valid comment. We asked participants to do a simple physical activity diary with the type and time spent every week. We added the details in the methods (lines 228-231) and results sections (lines 290-295).
- Intervention Fidelity
- Multiple instructors and settings are used (10 classes, 5 instructors), which may introduce variability.
- This should be acknowledged in the Limitations section.
Answer: we have added this point in the limitations section.
- Interpretation of Psychological Results
- The finding of increased external regulation is treated as a positive outcome. However, this is ambiguous without contextualisation.
Answer: Following the updated ANCOVA testing (vs previous t-test), there is now no significant findings regarding external regulation, but there was for amotivation which has been discussed in the Discussion section (starting line 495)
The self-determination theory (SDT) and associated three basic needs and motivation types has now been further outlined in the Introduction (lines 76-92) to give theoretical grounding. The decrease of amotivation has been contextualized in relation to the SDT in the Discussion.
- Recommend linking to Self-Determination Theory (Deci & Ryan) and reflecting critically on the motivational profile shift.
Answer: See above comment re SDT expansion in the Introduction and Discussion sections.
Statistical and Methodological Points
- The RAI (Relative Autonomy Index) is calculated but not discussed.
Answer: There was no significant findings for the RAI and this has been acknowledged in lines (from line 499) of the Discussion section in relation to how relatedness appears to have increased (support from increased Social Connectedness questionnaire scores and the qualitative themes of “belonging” and “relationships”) but autonomy and competence may not have increased.
- Normality testing is stated, but not reported. Suggest summarizing assumptions compliance.
Answer: we have added the p values for the Shapiro-Wilks test in the Statistical Analyses section (line 262).
- Balance outcomes may be affected by ceiling effects in the Berg Balance Scale. Suggest using more sensitive tools or reflecting this as a limitation.
- Answer: we have added this point in the limitations section: “our measure of balance had some limitations, with the Berg test showing a ceiling effect, which could have biased the results towards a lack of significant effect.”
Tables and Figures
- Tables 1–3 are dense and hard to interpret.
- Recommend adding at least one figure (e.g., STS or MCS scores over time) for visual clarity—this is common practice in MDPI journals.
Answer: we have added Figures 1 and 2 to show the significant effects on the STS and 6MWT. We have deleted these from Table 1, which should make it less dense.
References and Theoretical Framing
- Core literature on exercise motivation (e.g., Deci & Ryan 2000) is missing.
Suggest anchoring the BREQ-2 results in Self-Determination Theory to enhance conceptual rigour.
Answer: The self-determination theory (SDT) and associated three basic needs and motivation types has now been further outlined in the Introduction (lines 76-92) to give theoretical grounding. The decrease of amotivation has been contexualised in relation to the SDT in the Discussion.
Summary of Recommended Actions
- Substantial language and grammar revision throughout the manuscript.
- Strengthen the framing in the Introduction and Discussion, especially regarding novelty.
- Provide a more nuanced interpretation of psychological outcomes (especially motivation).
- Include at least one visualisation of the main results.
- Expand the theoretical discussion by integrating key frameworks from exercise psychology.
Answer: thanks again for your comments, we believe that we have answered them all, but do not hesitate to let us know if more needs to be done.
Reviewer 2 Report
Comments and Suggestions for Authors
Dear authors: congratulations on the quality of the work. Here are some suggestions to improve the manuscript (please focus on the main improvements related to statistical analysis and results).
Introduction:
The introduction could benefit from broadening the concept of SES and its impact on physical exercise and sports participation in this population (lines 58-65)
In line 72, What were the control groups to which the Zumba interventions were compared?
Procedures:
How were the groups randomized? ) (lines 118-121).
The type of activity performed by the control group should be described.
Pre-and Post-Testing Sessions: Physical Parameters
indicate the procedure for blood brachial pressure measurement and its reference (lines 126-130).
Zumba Gold® Intervention
With the measurement of absolute and relative heart rate, were decisions made about the intensity of the sessions?
Did the sessions include breaks between songs for active recovery?
Statistical analysis and Results
When using the t-test for between-group comparisons, there is a high risk of type 1 error. We recommend only using repeated-measures ANOVA and ƞp2 for effect sizes.
Discussion
the phrase (lines 401-403): “In addition, Zumba Gold includes relatively vigorous physical exercise (25.6% of class time was spent in the zone of peak HR in the present study), which is known to increase endorphin levels, even when estrogen concentrations are decreasing.” It could benefit from further discussion on how zumba gold can influence these physiological mechanisms.
In lines 431-437 In the BREQ-2 only “External regulation” is mentioned, without delving into self-determination theory (intrinsic vs. extrinsic motivation), nor how it might change over time. It is suggested to include further theoretical interpretation to show how the Zumba environment favors autonomy or competence.
Lines 344-447: A mean adherence of 65% is reported, but the reasons for this variability are not sufficiently explored. It is suggested that factors such as economic barriers, schedules, accessibility, or perceived self-efficacy in adherence be analyzed.
Limited interpretation of non-significant results
Lines 357-367 and 374-383
Lack of changes in balance or body composition are mentioned as “surprising” findings, but no in-depth reflection on possible reasons beyond the test (Berg).
Please strengthen this section with other factors such as training volume, age of the sample or duration of the intervention. The sensitivity of the instruments used and possible results with the gold standard (posturograph) could also be discussed.
conclusions
It is suggested to synthesize conclusions by variable or key findings. It becomes very long and confusing.
on lines 468-469, it is suggested to link the closing with non-significant findings.
Finally, it is suggested to add practical applications such as intensity (%FCMax) and effective session time to maximize sessions.
Author Response
Reviewer 2
Dear authors: congratulations on the quality of the work. Here are some suggestions to improve the manuscript (please focus on the main improvements related to statistical analysis and results).
Answer: thanks for your comments, we have addressed all your comments, and highlighted to changes in yellow in the text We have replied to your specific comments after each point (in red)
Introduction:
The introduction could benefit from broadening the concept of SES and its impact on physical exercise and sports participation in this population (lines 58-65)
Answer we have expanded this paragraph and added new references. See below (lines 65-75):
“This increased sedentariness is more important in older women from lower socioeconomic status who are twice as likely to be inactive than older women from high SES (SES, [8]). A study including 63 older adults including 40 women showed that neighbourhood SES was one of the main determinants of leisure-time physical activity in this population [9]. Despite these rather alarming results, older adults from low SES have not been the subject of many scientific studies on physical activity [10]. Interestingly, while older women from various SES share similar the barriers to physical activity, including proximity of sports facilities, physical symptoms, body image and intrinsic motivation linked to exercise type, some barriers, such as financial cost and accountability to others/social interactions have been shown to be specific to older women from lower SES [10-11].”
And lines 85-92: “A systematic review by Teixeira et al. (2012) reported consistent support for the positive relation between more autonomous forms of motivation to exercise, with a trend towards identified regulation predicting initial/short-term adoption more strongly than intrinsic motivation, and intrinsic motivation being more predictive of long-term exercise adherence. Indeed, a qualitative study assessing the motives and barriers to physical activity in older adults of different socioeconomic status highlighted the importance of integrated and identified regulations to promote physical activity engagement.”
In line 72, What were the control groups to which the Zumba interventions were compared?
Answer: this was added in the text (no exercise group), lines 100-101.
Procedures:
How were the groups randomized? ) (lines 118-121).
Answer: we used an online research randomizer (https://randomizer.org). This was added in the text, lines 134-135)
The type of activity performed by the control group should be described.
Answer: the time and type of physical activities performed by the control group were added, and some statistical analyses were done to see if there were any significant differences (it was not significant), lines 290-296.
Pre-and Post-Testing Sessions: Physical Parameters
indicate the procedure for blood brachial pressure measurement and its reference (lines 126-130).
Answer: This was added lines 154-158.
Zumba Gold® Intervention
With the measurement of absolute and relative heart rate, were decisions made about the intensity of the sessions?
Answer: no, HR was only measured to describe the intensity of the sessions, but nothing was done to change them.
Did the sessions include breaks between songs for active recovery?
Answer: only small water breaks were included, this was added in the text lines 242-243.
Statistical analysis and Results
When using the t-test for between-group comparisons, there is a high risk of type 1 error. We recommend only using repeated-measures ANOVA and ƞp2 for effect sizes.
Answer: we have changed our statistical analyses to ANCOVAs (due to another reviewer’s comments). Hence there are only partial eta squared reported now, and we believe, less chance of type 1 error.
Discussion
the phrase (lines 401-403): “In addition, Zumba Gold includes relatively vigorous physical exercise (25.6% of class time was spent in the zone of peak HR in the present study), which is known to increase endorphin levels, even when estrogen concentrations are decreasing.” It could benefit from further discussion on how zumba gold can influence these physiological mechanisms.
Answer: we added some explanation to this part of the discussion, including the specific movements considered as high intensity in Zumba and a sentence to explain the mechanisms further (lines 462-466): “In addition, Zumba Gold® includes relatively vigorous physical exercise (25.6% of class time spent in the vigorous to maximal HR zone in the present study), due to its relatively fast pace, involvement of major muscle groups of the upper limb, lower limb and trunk and frequent changes of direction. Vigorous exercise is known to increase endorphin levels, even when estrogen concentrations are decreasing [52]. This is achieved by an activation of the body’s stress response, mainly through the hypothalamic-pituitary-adrenal axis, leading to the release of endorphins [52].”
In lines 431-437 In the BREQ-2 only “External regulation” is mentioned, without delving into self-determination theory (intrinsic vs. extrinsic motivation), nor how it might change over time. It is suggested to include further theoretical interpretation to show how the Zumba environment favors autonomy or competence.
Answer: Following the updated ANCOVA testing (vs previous t-test), there is now no significant findings regarding external regulation, but there was for amotivation which has been discussed in the Discussion section (starting line 495)
The SDT and associated three basic needs and motivation types has now been further outlined in the Introduction (lines 76-92) to give theoretical grounding. The decrease of amotivation has been contextualized in relation to the SDT in the Discussion.
Lines 344-447: A mean adherence of 65% is reported, but the reasons for this variability are not sufficiently explored. It is suggested that factors such as economic barriers, schedules, accessibility, or perceived self-efficacy in adherence be analyzed.
Answer: we added more about this in two places in the discussion:
-First, the first time we mentioned the adherence and large standard deviations, we mentioned the factors that you suggested, and others, linking them to the papers described in our introduction (line 388-392).
-Further one, we also added some explanation linked to the specific motivations of women from low SES: “Additionally, the adherence rates may have also limited the potential for basic needs development. Indeed, a study by Bukan et al. [59] compared focus group responses from high and low SES and reported that low SES individuals seemed to be especially motivated to change their lifestyle when they experienced health complaints, but were rather hesitant to change their lifestyle for preventive purposes. This may reflect the lower levels of amotivation in the current ZG group but the lack of increase in the other more adaptive forms of motivation. It should also be noted that the ZG group had relatively low levels of amotivation at baseline. Nevertheless, Bukan et al. [59] also reported that individuals in the low SES group preferred physical activities where groups consisted of individuals of the same age, gender and physical condition. These characteristics were certainly evident in the current exercise programme. “
Limited interpretation of non-significant results
Lines 357-367 and 374-383
Lack of changes in balance or body composition are mentioned as “surprising” findings, but no in-depth reflection on possible reasons beyond the test (Berg).
Please strengthen this section with other factors such as training volume, age of the sample or duration of the intervention. The sensitivity of the instruments used and possible results with the gold standard (posturograph) could also be discussed.
Answer: Regarding the lack of change of body composition, we added some detail about the factors that you mentioned as well as the results of two new studies lines 405-415 “the limited duration, frequency and training volume, and other factors such as our participants’ age and their diet may have influenced this non significant result for body composition. For example, Barbalho et al. [48] compared the effects of a 12-week resistance training programme at low vs. high volume in healthy older women and showed greater decreases in body weight and waist circumference in the high volume group. In addition, age is a well-known factor to impede weight loss in middle-aged and older women, mainly due to the effects of the menopause on muscle mass and metabolism [3-4]. Finally, our intervention did not include any modification of our participants’ diet, which could further explain the lack of significant changes on body fat in the ZG group. Indeed, many studies reported the benefits of calorie restriction in addition to physical exercise on weight loss compared to exercise alone [49].”
Regarding the lack of change in balance, we already discussed the limitation of the Berg test, so we just added a comment about the Gold standard lines 432-434.
Conclusions
It is suggested to synthesize conclusions by variable or key findings. It becomes very long and confusing.
on lines 468-469, it is suggested to link the closing with non-significant findings.
Finally, it is suggested to add practical applications such as intensity (%FCMax) and effective session time to maximize sessions.
Answer: we have shortened the introduction and grouped results by types (physical/psychological), and also added the non significant results as well as the practical applications.
Reviewer 3 Report
Comments and Suggestions for Authors
The authors need to address the following comments for its possible publication.
The authors used many variables in this study. However, the introduction does not sufficiently explain why these variables are important. The introduction should justify the importance and necessity of the study variables.
The power calculation is based on a different study population and endpoint, raising questions about its validity. Given multiple endpoints (physical, psychological, qualitative), the study risks inflated Type I error due to multiple comparisons with no adjustment. Some reported significant results have small effect sizes (ƞp² < 0.15), suggesting marginal practical significance even if statistically significant.
The presence of statistically significant baseline differences (e.g., HADS anxiety, BREQ-2 Amotivation, BREQ-2 external regulation, and Y-balance values) suggests that randomization did not achieve equivalence between groups. The current repeated-measures ANOVA without adjusting for baseline scores risks attributing pre-existing differences to the intervention. The authors may use ANCOVA or mixed models with baseline values as covariates to correctly isolate intervention effects.
The finding of increased external regulation (BREQ-2) is interpreted as positive via group motivation. However, external regulation is generally considered less desirable for long-term adherence compared to intrinsic/identified regulation. Authors should discuss this nuance, particularly in contrast to prior studies showing gains in intrinsic motivation. And The authors must thoroughly review Self-Determination Theory, paying particular attention to internalization and the impact of autonomy.
The discussion correctly identifies ceiling effects in Berg Balance Scale, but this limitation undermines one of the stated aims (balance improvement). The authors may reframe conclusions to avoid implying that Zumba Gold® has “no effect” on balance without more sensitive measures.
No significant change in fat mass or muscle mass; however, this contradicts some prior studies. The authors attribute this to frequency and participants’ BMI. However, the diet variable is uncontrolled—this is a major confounder and should be acknowledged explicitly.
While thematic analysis is a strength, coding procedures and inter-rater reliability are not reported. This weakens trust in qualitative findings. I would recommend adding triangulation details.
"Zumba Gold®" should be consistently styled throughout (occasionally appears without ®.
Some tables (e.g., Table 1) have inconsistent decimal places. Abbreviations should be defined at first appearance in each table.
Some citations are quite recent and strengthen novelty; however, inclusion of more critical work on motivation theory in older adults (Deci & Ryan, Self Determination Theory framework) would be beneficial.
Author Response
The authors need to address the following comments for its possible publication.
Answer: thanks for your comments, we have addressed all your comments, and highlighted to changes in yellow in the text We have replied to your specific comments after each point (in red)
The authors used many variables in this study. However, the introduction does not sufficiently explain why these variables are important. The introduction should justify the importance and necessity of the study variables.
Answer: thanks for your valuable comment. We have justified the use of each type of variable in the introduction, lines 53-57, and later on as well. This section has also been partly rewritten due to the comments of another reviewer. To summarise, we tried to link our variables to some aspects of mental health/fitness of post-menopausal women, as shown below:
-Body weight,body fat: increase in fat after the menopause
-STS: decrease lower body strength and sarcopenia in older adults
-blood pressure, 6MWT: decreased cardiovascular fit ness in older adults and benefits of Zumba on these.
-Balance: loss of balance due to sarcopenia, in particular.
-BREQ2: lines 76-92.
Other psychological variables (anxiety, depression, fatigue, etc…) are justified by the expected effect of Zumba (lines 94-110).
The power calculation is based on a different study population and endpoint, raising questions about its validity. Given multiple endpoints (physical, psychological, qualitative), the study risks inflated Type I error due to multiple comparisons with no adjustment. Some reported significant results have small effect sizes (ƞp² < 0.15), suggesting marginal practical significance even if statistically significant.
Answer: It was really difficult to find a suitable study for the power analysis, so we have decided to perform an a priori power calculation, with an effect size of 0.5, an alpha level of 0.05 and a power=0.8 (lines 124-126). It suggested a sample size of 34. We also changed our statistical analyses to ANCOVAs due to your next comment and the comment from another reviewer. This means that we only have partial eta squared and no multiple comparisons, hence decreasing the risk of type 1 error. The small effect sizes were highlighted in the discussion.
The presence of statistically significant baseline differences (e.g., HADS anxiety, BREQ-2 Amotivation, BREQ-2 external regulation, and Y-balance values) suggests that randomization did not achieve equivalence between groups. The current repeated-measures ANOVA without adjusting for baseline scores risks attributing pre-existing differences to the intervention. The authors may use ANCOVA or mixed models with baseline values as covariates to correctly isolate intervention effects.
Answer: Thanks for this valuable comment, we have performed ANCOVAs with baseline data as covariates, see lines 266-270: “an analysis of covariance (ANCOVA) was performed on all outcomes variables to assess the effects of group (ZG vs. C) on the post-intervention data, with the baseline data as a covariate. Effect sizes were calculated as Partial Eta Squared (ƞp2) and interpreted as no effect (0-0.05), minimum effect (0.05-0.26), and strong effect (0.26-0.64), [40].”
It has changed a few of the results, but not many (the Amotivation item of the BREQ2 and muscle mass are now significantly different between groups, and the External regulation of the BREQ2 is not significant anymore). We changed the results and discussion of these items accordingly.
The finding of increased external regulation (BREQ-2) is interpreted as positive via group motivation. However, external regulation is generally considered less desirable for long-term adherence compared to intrinsic/identified regulation. Authors should discuss this nuance, particularly in contrast to prior studies showing gains in intrinsic motivation. And The authors must thoroughly review Self-Determination Theory, paying particular attention to internalization and the impact of autonomy.
Answer: Following the updated ANCOVA testing (vs previous t-test), there is now no significant findings regarding external regulation, but there was for amotivation which has been discussed in the Discussion section (starting line 496)
The SDT and associated three basic needs and motivation types has now been further outlined in the Introduction (lines 76-92) to give theoretical grounding. The decrease of amotivation has been contextualized in relation to the SDT in the Discussion.
The discussion correctly identifies ceiling effects in Berg Balance Scale, but this limitation undermines one of the stated aims (balance improvement). The authors may reframe conclusions to avoid implying that Zumba Gold® has “no effect” on balance without more sensitive measures.
Answer: we added a sentence at the end of this part to take your comment into account (lines 432-434): “It is possible that we would have observed a significant difference between groups if we had used a more sensitive measure.”
No significant change in fat mass or muscle mass; however, this contradicts some prior studies. The authors attribute this to frequency and participants’ BMI. However, the diet variable is uncontrolled—this is a major confounder and should be acknowledged explicitly.
Answer: The effect of diet (as well as other factors requested by another reviewer) were added lines 405-415: “Finally, our intervention did not include any control or modification of our participants’ diet, which could further explain the lack of significant changes on body fat in the ZG group. Indeed, many studies reported the benefits of calorie restriction in addition to physical exercise on weight loss compared to exercise alone [49]”.
While thematic analysis is a strength, coding procedures and inter-rater reliability are not reported. This weakens trust in qualitative findings. I would recommend adding triangulation details.
Answer: Greater detail regarding the qualitative analysis has been added to lines 273-287 of the transcript. This outlines the use of 2 authors to ensure accuracy of the transcripts as well as consideration of confirmability, reflexivity and sincerity. This was supported by a second author acting as a critical friend to encourage reflection and exploration of alternative interpretations.
"Zumba Gold®" should be consistently styled throughout (occasionally appears without ®.
Answer: we checked the manuscript throughout and corrected this.
Some tables (e.g., Table 1) have inconsistent decimal places. Abbreviations should be defined at first appearance in each table.
Answer: We have used a number of decimal places used depending on the number (smaller number has more decimal places, such as the BREG-2 amotivation item) and precision needed (i.e., blood pressure no decimal place by body weight one decimal place). Let us know if you would like us to change this. We do not believe that the number of decimal places has to be consistent within tables. We have defined some abbreviations in the titles, where missing.
Some citations are quite recent and strengthen novelty; however, inclusion of more critical work on motivation theory in older adults (Deci & Ryan, Self Determination Theory framework) would be beneficial.
Answer: As above, the SDT and associated three basic needs and motivation types has now been further outlined in the Introduction to give theoretical grounding. The decrease of amotivation has been contexualised in relation to the SDT in the Discussion (starting line 499).
Consideration of this has been placed in the Introduction (see lines 75-91) and also in the Discussion section.
Specific literature related to motivational profiles different SES groups has been outlined in the Introduction:
-lines 90-92: (Gray PM, Murphy MH, Gallagher AM, Simpson EE. Motives and barriers to physical activity among older adults of different socioeconomic status. Journal of Aging and Physical Activity. 2016 Jul 1;24(3):419-29.)
-Health perceptions of low SES individuals (Bukman AJ, Teuscher D, Feskens EJ, van Baak MA, Meershoek A, Renes RJ. Perceptions on healthy eating, physical activity and lifestyle advice: opportunities for adapting lifestyle interventions to individuals with low socioeconomic status. BMC Public Health. 2014 Oct 4;14:1036. doi: 10.1186/1471-2458-14-1036. PMID: 25280579; PMCID: PMC4210550) has also been discussed in the Discussion section.
Reviewer 4 Report
Comments and Suggestions for Authors
First of all, I would like to thank you for the opportunity to review this article, which is of considerable interest and holds significant potential for improving the quality of life of individuals with low socioeconomic status.
Below are my comments to clarify certain aspects.
It would be very useful to clarify the method used to assess the level of physical activity, which served as the basis for the sample selection.
The reference to body mass index is missing from Table 1.
I would recommend modifying the title to refer to “older women” rather than “postmenopausal women,” as this topic is neither discussed in the introduction nor the discussion, and the majority of the sample consists of women over 60 years of age, indicating that they have already been in menopause for nearly a decade.
It would also be helpful to specify the activities undertaken by the control group, since Table 1 shows an improvement in balance for this group, while the experimental group shows a decrease, although this difference is not statistically significant.
Author Response
Reviewer 4
First of all, I would like to thank you for the opportunity to review this article, which is of considerable interest and holds significant potential for improving the quality of life of individuals with low socioeconomic status.
Below are my comments to clarify certain aspects.
Answer: thanks for your comments, we have addressed all your comments, and highlighted to changes in yellow in the text We have replied to your specific comments after each point (in red)
It would be very useful to clarify the method used to assess the level of physical activity, which served as the basis for the sample selection.
Answer: this was added in the text line 129-130. We only used a simple diary with type of PA and time spent. Some detail were also added in the results section.
The reference to body mass index is missing from Table 1.
Answer: we did not report BMI as it is less valid than body fat to assess overweight and obesity.
I would recommend modifying the title to refer to “older women” rather than “postmenopausal women,” as this topic is neither discussed in the introduction nor the discussion, and the majority of the sample consists of women over 60 years of age, indicating that they have already been in menopause for nearly a decade.
Answer: Since no other reviewer has requested this change, and since we address menopausal changes in the introduction, we believe that it should stay in the title. Indeed, most of the changes we are trying to address with Zumba are dur to the menopause, even if our participants are older. Please let us know if you would like us to reconsider.
It would also be helpful to specify the activities undertaken by the control group, since Table 1 shows an improvement in balance for this group, while the experimental group shows a decrease, although this difference is not statistically significant.
Answer: we have added these in the results section lines 290-295.
Reviewer 5 Report
Comments and Suggestions for Authors
This randomized controlled trial investigates the physical and psychological effects of a 12-week Zumba Gold® (ZG) intervention in sedentary postmenopausal women from low socioeconomic status (SES). Using a mixed-methods approach (quantitative assessments + qualitative interviews), the authors analyzed 43 participants (23 ZG, 20 control). Key findings include: (1) Significant improvements in lower limb strength (sit-to-stand test: +9.0%) and cardiorespiratory endurance (6-min walk test: +4.5%); (2) Enhanced mental health (SF-12 MCS: +15.7%), reduced fatigue (MFI-20 general/mental fatigue: -12.4%/-13.8%), and increased social connectedness (+9.8%) exclusively in the ZG group; (3) Higher external motivation (BREQ-2 external regulation: +200%); and (4) Qualitative themes (belonging, non-judgmental environment, psychological motivation, mind-body connection) contextualizing benefits. The study addresses a critical gap by targeting an understudied, high-risk population.
Major Points
1. Significant baseline differences existed in anxiety (HADS-A: ZG 4.74 vs. C 6.95; p=0.017) and external motivation (BREQ-2: ZG 0.13 vs. C 0.38; p=0.048). These were not adjusted for in analyses, potentially confounding post-intervention group comparisons.
2. Increased external (not intrinsic) motivation in ZG is framed negatively (Lines 414-415). For low-SES populations, social accountability (e.g., group commitment) may be a key adherence driver – reframe as a context-appropriate strength.
Minor Points
3. Table 1: Missing standard deviations for "difference" 95% CIs (e.g., STS: "0.3 to 2.3" lacks ±SD). Report full descriptive statistics per CONSORT.
4. Attendance was 65±25% (Line 206). High SD suggests financial/time barriers; correlate compliance with outcomes (e.g., dose-response).
5. Scale ceiling effects (baseline: 54.8/56; Line 228) and Y-balance’s strength dependence (Line 388) likely obscured true effects. Use force platforms in future studies.
6. Non-significant fat increase in ZG (35.3%→36.2%; Table 1) warrants comment given menopausal metabolic risks.
7. Repository link (Line 519) states "not report any data" – clarify accessibility.
8. For ANOVA, report exact p-values (e.g., STS p=0.012) instead of asterisks in tables (e.g., Table 1 STS "**").
9. Only 25.6% exercise time was in vigorous zones (Line 228). Align interpretation with ACSM’s moderate-intensity focus (Line 366).
10. No objective monitoring (e.g., accelerometry) was performed for the control group activity – risk of contamination if controls increased PA.
Author Response
Reviewer 5
This randomized controlled trial investigates the physical and psychological effects of a 12-week Zumba Gold® (ZG) intervention in sedentary postmenopausal women from low socioeconomic status (SES). Using a mixed-methods approach (quantitative assessments + qualitative interviews), the authors analyzed 43 participants (23 ZG, 20 control). Key findings include: (1) Significant improvements in lower limb strength (sit-to-stand test: +9.0%) and cardiorespiratory endurance (6-min walk test: +4.5%); (2) Enhanced mental health (SF-12 MCS: +15.7%), reduced fatigue (MFI-20 general/mental fatigue: -12.4%/-13.8%), and increased social connectedness (+9.8%) exclusively in the ZG group; (3) Higher external motivation (BREQ-2 external regulation: +200%); and (4) Qualitative themes (belonging, non-judgmental environment, psychological motivation, mind-body connection) contextualizing benefits. The study addresses a critical gap by targeting an understudied, high-risk population.
Major Points
Answer: thanks for your comments, we have addressed all your comments, and highlighted to changes in yellow in the text We have replied to your specific comments after each point (in red)
Significant baseline differences existed in anxiety (HADS-A: ZG 4.74 vs. C 6.95; p=0.017) and external motivation (BREQ-2: ZG 0.13 vs. C 0.38; p=0.048). These were not adjusted for in analyses, potentially confounding post-intervention group comparisons.
We have changed our statistical analyses to ANCOVAs due to the comments from another reviewer. Our ANCOVAs used the baseline data as covariate, and the post-test data as main variable. This means that we only have partial eta squared and no multiple comparisons, hence decreasing the risk of type 1 error.
Lines 266-270: “Consequently, an analysis of covariance (ANCOVA) was performed on all outcomes variables to assess the effects of group (ZG vs. C) on the post-intervention data, with the baseline data as a covariate. Effect sizes were calculated as Partial Eta Squared (ƞp2) and interpreted as no effect (0-0.05), minimum effect (0.05-0.26), and strong effect (0.26-0.64), [40].”
Increased external (not intrinsic) motivation in ZG is framed negatively (Lines 414-415). For low-SES populations, social accountability (e.g., group commitment) may be a key adherence driver – reframe as a context-appropriate strength.
Following the updated ANCOVA testing (vs previous t-test), there is now no significant findings regarding external regulation, but there was for amotivation which has been discussed in the Discussion section (starting line 496)
In particular, the reasons for the absence of an increase in the different types of extrinsic and intrinsic motivation have been discussed in relation to contrasting other studies and the specific contextual barriers of low SES groups. Additionally, literature is utilized to outline how the current intervention does encompass key preferential characteristics (similar age, group and fitness level) as suggested from focus groups with low SES groups.
Minor Points
3. Table 1: Missing standard deviations for "difference" 95% CIs (e.g., STS: "0.3 to 2.3" lacks ±SD). Report full descriptive statistics per CONSORT.
Answer: we added SD for the 95% CI into brackets next to each variable in all tables.
Attendance was 65±25% (Line 206). High SD suggests financial/time barriers; correlate compliance with outcomes (e.g., dose-response).
Answer: we added some discussion to the large SD observed in attendance lines… “In contrast, the relatively high variability in adherence could be explained by socio-economic factors (financial constraints, integration of exercise in working and daily life), social factors (lack of support network) and psychological factors that affect the perceived self-efficacy of women from low SES (negative body image, fear of judgement), [9-11].”
Regarding the correlation, it would not be valid to correlate compliance to post-intervention data, as it would not take into account the baseline data that we used as a covariate. It might be better to correlate compliance with the delta (change between baseline and post) but we are concerned that it would add some confusion for the reader, as we did not mention the delta in the rest of the results. In addition, other reviewers mentioned that we have a lot of data/statistical analyses and advised to streamline the results. We have decided to not do this for now, but if you believe that we should do this analysis, then we are happy to do so in the next review step, but wanted to check with you first, given our previous comments.
Scale ceiling effects (baseline: 54.8/56; Line 228) and Y-balance’s strength dependence (Line 388) likely obscured true effects. Use force platforms in future studies.
Answer: Our statistical analyses were changed to ANCOVAs, with baseline data as a covariate. We believe this is now better scaling the ceiling effects to the baseline differences. Regarding the Y balance test, we added a comment on its reliance on strength in the discussion lines… “We also used the Y balance test that may not allow people with some musculoskeletal issues or a lack of strength to reach a certain distance, despite good balance [50].”
Non-significant fat increase in ZG (35.3%→36.2%; Table 1) warrants comment given menopausal metabolic risks.
Answer: we added a comment on this in the discussion line… “It should be noted, however, that the ZG increased their body fat between baseline and post-intervention, which could lead to greater metabolic risk, although this change was non significant.”
- Repository link (Line 519) states "not report any data" – clarify accessibility.
Answer: Thanks for pointing this out, we have removed this part and the link now works.
For ANOVA, report exact p-values (e.g., STS p=0.012) instead of asterisks in tables (e.g., Table 1 STS "**").
Answer: we do not believed that it is common practice to do this in this journal, and it could make the tables and graphs messy. We will check with the editorial office and correct at the nest revision step if needed.
Only 25.6% exercise time was in vigorous zones (Line 228). Align interpretation with ACSM’s moderate-intensity focus (Line 366).
Answer: we have added a comment in the discussion lines “In addition, Zumba Gold® includes relatively vigorous physical exercise (25.6% of class time (about 15-min) spent in the vigorous to maximal HR zone in the present study, which is in line with the ACSM recommendations to spend 75-min weekly performing vigorous-intensity exercise if this was repeated daily [39]).”
No objective monitoring (e.g., accelerometry) was performed for the control group activity – risk of contamination if controls increased PA.
Answer: We only monitored physical activity with a diary, where participants recorded the type and time spent doing PA. WE have reported these in the methods and results, and no significant difference was seen between groups, however we are aware that it is a less valid method to record PA, and hence we have added this a limitation in the discussion.
Round 2
Reviewer 1 Report
Comments and Suggestions for Authors
I accept the answers and coreection version.
Author Response
Thank you so much for accepting our corrections
Reviewer 2 Report
Comments and Suggestions for Authors
Introduction
The SES concept is incorporated in line "This increased sedentariness is more important in older women from lower socioeconomic status who are twice as likely to be inactive than older women from high SES. Please define the abbreviation beforehand.
Zumba gold intervention
If the control group was described as ‘inactive’ but also as ‘engaging in physical activity,’ does this not contradict the first inclusion criterion for participants? I suggest clarifying that the control group engaged in physical activity but did not meet the minimum weekly physical activity requirements
Limitations
I suggest that, in the limitations section, it should be mentioned that the control group did not take part in any other traditional planned exercise program.
Conclusions
I suggest indicating that future research should compare Zumba Gold with other forms of planned exercise (e.g., HIIT, concurrent training, etc.)
Author Response
Reviewer 2
The SES concept is incorporated in line "This increased sedentariness is more important in older women from lower socioeconomic status who are twice as likely to be inactive than older women from high SES. Please define the abbreviation beforehand.
Answer: we have added the abbreviation in this sentence to define it: “This increased sedentariness is more important in older women from lower socioeconomic status (SES) who are twice as likely to be inactive than older women from high SES [8].”
Zumba gold intervention
If the control group was described as ‘inactive’ but also as ‘engaging in physical activity,’ does this not contradict the first inclusion criterion for participants? I suggest clarifying that the control group engaged in physical activity but did not meet the minimum weekly physical activity requirements.
Answer: We removed the term “sedentary” in our inclusion criteria and only kept the reference to the PA guidelines: “Inclusion criteria were not meeting the guidelines for physical activity for this age group (at least 150 minutes of weekly moderate intensity activity or 75 minutes of vigorous intensity, www.nhs.uk) as assessed with a physical activity diary.”
Limitations
I suggest that, in the limitations section, it should be mentioned that the control group did not take part in any other traditional planned exercise program.
Answer: this was added in the limitation section: “Another potential limitation is that we compared or ZG group to individuals who did not take part in any structured exercise program. It could be interesting to use a comparison group involved in another type of exercise, such as high-intensity interval training (HIIT), for example.”
Conclusions
I suggest indicating that future research should compare Zumba Gold with other forms of planned exercise (e.g., HIIT, concurrent training, etc.)
Answer: this was added in the discussion: “It could be interesting, in particular, to compare Zumba Gold® to other types of planned exercise (HIIT, concurrent training, etc…).”
Reviewer 3 Report
Comments and Suggestions for Authors
The authors have addressed the commments, and the manuscript has been improved sufficiently for its possible publication. One minor thing is that the manuscript used the template of 'biology', and it needs to be changed to 'healthcare' template. Thank the authors for their efforts on this manuscript.
Author Response
Thank you for accepting our corrections. I think we can change the format to Healthcare format at the next step, we will ask the editor.